# Developmental compartments in the larval trachea of Drosophila

**Prashanth R Rao[†], Li Lin[†‡], Hai Huang, Arjun Guha[§¶], Sougata Roy[**],
Thomas B Kornberg[*]**

Cardiovascular Research Institute, University of California, San Francisco, San Francisco, United States

**\*For correspondence:**
tkornberg@ucsf.edu

[†]These authors contributed equally to this work

**Present address:** [‡]PharmaLex GmbH, Mannheim, Germany; [§]Pulmonary Center, Boston University School of Medicine, Boston University, Boston, United States; [¶]Institute for Stem Cell Biology and Regenerative Medicine, National Centre for Biological Sciences, Bangalore, India; [**]Department of Molecular and Cellular Biology, University of Maryland, College Park, United States

**Competing interests:** The authors declare that no competing interests exist.

**Abstract** The Drosophila tracheal system is a branched tubular network that forms in the embryo by a post-mitotic program of morphogenesis. In third instar larvae (L3), cells constituting the second tracheal metamere (Tr2) reenter the cell cycle. Clonal analysis of L3 Tr2 revealed that dividing cells in the dorsal trunk, dorsal branch and transverse connective branches respect lineage restriction boundaries near branch junctions. These boundaries corresponded to domains of gene expression, for example where cells expressing Spalt, Delta and Serrate in the dorsal trunk meet *vein*–expressing cells in the dorsal branch or transverse connective. Notch signaling was activated to one side of these borders and was required for the identity, specializations and segregation of border cells. These findings suggest that Tr2 is comprised of developmental compartments and that developmental compartments are an organizational feature relevant to branched tubular networks.

## Introduction

The importance of temporal- and position-specific gene expression to regional specialization is established for many tissues. It may be universal. In contrast, the roles and contributions of determined cell lineages are not as clear and may not be generally shared. In Drosophila, there are tissues that are dependent on cell lineages - examples include neurons in the developing brain (reviewed in *Spindler and Hartenstein, 2010*) and the developmental compartments that produce the adult cuticle (reviewed in *Crick and Lawrence, 1975*). However, there are also organs such as the salivary gland, gut and tracheal system of the Drosophila embryo that form from groups of non-mitotic cells (reviewed in *Kerman et al., 2006*). The cells of these organs are assigned to their roles after they have exited the cell cycle, and the programs of tubulogenesis and branching that generate them proceed without further cell divisions. Even the complex network of tracheal branches in the embryo forms in this way, and without an apparent role for defined lineages (*Samakovlis et al., 1996*).

The tracheal primordia, which have 80–90 cells, have been estimated to derive from approximately ten blastoderm-stage cells (*Campos-Ortega and Hartenstein, 1985*). The primordia are first evident in the lateral embryonic ectoderm during stage 10 (4–5 hr after egg laying (AEL)) as twenty groups of cells with distinct morphology, and in early in stage 11, they form discrete pits (*Wilk et al., 1996*). Three post-blastoderm cell divisions have occurred by early stage 11, but there are no additional mitoses until L3, when some tracheal cells re-initiate cell cycling (*Guha and Kornberg, 2005*; *Guha et al., 2008*; *Sato and Kornberg, 2002*; *Weaver and Krasnow, 2008*).

Beginning at embryo stage 11, the tracheal pits invaginate, elongate and mold into more complex shapes, following a stereotyped program that generates the major tracheal branches (*Samakovlis et al., 1996*). Although analysis of random clones revealed no consistent relationships between position and kinship that would suggest a role for cell lineage in assigning cells to a particular tracheal branch (*Samakovlis et al., 1996*), there is evidence for region-specific and stage-specific

**eLife digest** As a fruit fly develops, its cells may sort themselves into groups according to the type of cell that they will eventually become. Some groups form 'developmental compartments' that are separated by boundaries that cells cannot move across. All the descendants of a cell in a compartment will activate the same specific gene (called a 'selector' gene) that determines their identity and fate. Similar compartments also form in the developing hindbrains of mammals, but it is not clear how general this mechanism of tissue patterning is.

Fruit fly larvae undergo a physical transformation called metamorphosis to become adult fruit flies. Here, Rao et al. discover that the cells in the developing airways (or trachea) of the larvae at the start of metamorphosis are organised into compartments. At this stage the cells in the trachea start to divide and grow to make the adult tracheal system. The experiments show that these cells do not spread from one main branch of the tracheal system into another. Instead, the cells cluster in locations where the different branches meet on either side of a straight boundary.

The cells on each side of these boundaries activate different genes that regulate their identity and development. For example, cells in one branch of the system switch on a selector gene that makes a protein called Spalt. A pathway known as Notch signaling is activated by cells on the other side of a nearby boundary in a different branch of the tracheal system. This separation of Spalt production and Notch activation establishes a cell communication system that keeps the cells of the different compartments apart.

Rao et al.'s findings reveal a role for the Notch protein in regulating the organization of cells into compartments to form branches in fruit fly airways. A future challenge is to find out if Notch plays a similar role in other branched tissues, such as blood vessels.

gene expression, and mutant phenotypes suggest that these genes have essential roles in the morphogenetic processes that generate the branches. For example, *tracheless (trh)* is expressed by pit cells and in all tracheal progenitors, and in *trh* mutants, pits do not form and there is no apparent tracheal development (*Isaac and Andrew, 1996*; *Wilk et al., 1996*). *Spalt (sal)* is expressed in the dorsal part of the tracheal primordium that will form the dorsal branch (DB) and dorsal trunk (DT), and in *sal* mutants, the dorsal primordium expresses genes such as *unplugged,* which is normally only expressed ventrally, and its cells do not migrate normally (*Franch-Marro and Casanova, 2002*). These and other findings have been interpreted as dorsal to ventral transformations and as evidence that *sal* has a role in fate specification for particular branches (*Chen et al., 1998*; *Kuhnlein and Schuh, 1996*). Mutants defective for *knirps (kni)* and for Notch signaling have major branching abnormalities suggestive of general and persistent requirements that begin at the earliest primary branching stages (*Chen et al., 1998*; *Ghabrial and Krasnow, 2006*; *Ikeya and Hayashi, 1999*; *Lli-margas, 1999*; *Steneberg et al., 1999*). Although these studies support the idea that specialization and branch formation are dependent on region-specific expression of several fate-determining genes, the expression patterns of these genes have not been precisely correlated (at cellular resolution) with branching morphologies. There is evidence supporting the presence of a *kni*-dependent border between DT and DB cells in the embryo (*Chen et al., 1998*), but it is not known whether the different branches, either alone or in combination, exist as distinct regions that develop with unique genetic addresses.

Developmental compartments are regions whose constituent cells share a unique genetic address and are clonally isolated. They are polyclones (*Crick and Lawrence, 1975*) - groups of cells that represent all the descendants of a small group of founder cells that grow but never mix with cells of other compartments or tissues. Compartments of the imaginal discs and abdominal histoblasts generate the epidermal and neuronal cells of the adult cuticle (*Chen and Baker, 1997*; *Garcia-Bellido et al., 1973*; *Kornberg, 1981a*; *Morata and Lawrence, 1978*, *1979*). They do not include associated cells, such as the myoblasts and tracheal cells that develop together and in tight juxtaposition with the epithelial cells of the wing imaginal disc. The common ancestry of cells in the developmental compartments in the epithelia of imaginal discs is essential to the identity and function of these domains, and the adult cuticle does not develop normally if the compartment boundaries do not restrict cells to grow on one side or other (*Blair et al., 1994*; *Diaz-Benjumea and Cohen, 1993*;

*Kornberg, 1981a, b*; *Morata and Lawrence, 1975*), or do not properly delimit the expression of certain genes to the cells of a particular compartment (*Dominguez et al., 1996*; *Tabata et al., 1995*). The compartments are domains of gene expression and pattern whose geographical positions and limits are precisely defined. In the wing disc, the compartment borders set up and coincide in space with developmental organizers (*Diaz-Benjumea and Cohen, 1993*; *Tabata et al., 1995*) and juxtapose groups of cells with opposite developmental polarity (*Chen and Struhl, 1996*; *Chuang and Kornberg, 2000*; *Garcia-Bellido and Santamaria, 1972*; *Lawrence and Morata, 1976*; *Lawrence et al., 2007*; *Tabata et al., 1995*), but the generality of developmental compartments beyond the epithelial progenitors of the insect cuticle is uncertain. It is not known whether cell lineage domains that have been identified in other tissues, for example in the adult thoracic muscles (*Lawrence, 1982*) and midgut (*Marianes and Spradling, 2013*), share these properties.

The work described here investigated cell lineage parameters in the Drosophila tracheal system at the L3 stage. In contrast to the process that forms the tracheal system in the embryo, the pupal and adult tracheal systems develop as their constituent cells proliferate. As noted above, the first post-embryonic cell divisions in Tr2 occur during L3 when its branches begin to reorganize in preparation for metamorphosis. We undertook a classical clonal analysis of the branches of Tr2 and found evidence of coincident lineage and gene expression domains.

## Results

### The dynamics of tracheoblast proliferation in the dorsal trunk

At the beginning of L3, the branches of Tr2 are sparsely populated by large cells (*Figure 1A*). The DT has 16–18, the DB 8, and the transverse connective (TC) 5–7 (*Guha and Kornberg, 2005*; *Guha et al., 2008*; *Lin, 2009*; *Weaver and Krasnow, 2008*); there is no air sac primordium (ASP). The other tracheal metameres are similarly constituted, but in early L3, programs of cell division that are unique to Tr2 repopulate its branches with many cells and grow the ASP (*Figure 1A,B*) (*Guha and Kornberg, 2005*; *Sato et al., 2008*). At late L3, the DT has approximately 360 ± 40 cells (*Lin, 2009*). The DB also reinitiates cell cycling during L3 (*Weaver and Krasnow, 2008*). Despite the difference in cell numbers between Tr2 and the other metameres, the general overall size of the branches in each metamere remains similar as the L3 larva grows because the proliferating Tr2 cells are smaller. To better understand the growth dynamics of the Tr2 tracheoblasts during L3, we induced random clones of marked cells in the DT and analyzed their size and distribution.

We induced recombination in late embryos (tracheoblasts do not divide between embryo stage 11 and >10 hr after the L2 to L3 molt) and examined 1680 tracheal preparations for marked DT cells at various times during the L3 period. 185 marked patches were identified. At 0–2 hr after the L2-3 molt, 22/24 marked cells appeared to be isolated singles without marked neighbors, as would be expected prior to onset of proliferation and for the observed frequency of recombination (11%). At 16–18 hr , 80% were still singles, but at 18–20 hr, 69% were clones of two adjacent cells. This suggests that re-entry into the cell cycle was synchronous and that most DT cells divide according to a similar schedule. The clones formed contiguous patches (*Figure 1C*), indicating that the DT cells do not tend to migrate or intermix with their neighbors; there was no apparent bias to their distribution within the DT. Divisions continued with approximate synchrony and with cycle time (at 23°C) of 10–12 hr (*Figure 1B*). The time interval during L3 that follows the onset of mitotic cycling can therefore accommodate five divisions, and if all of the starting population contributes to growth, the five divisions are sufficient to produce all the cells in the DT of the wandering L3. The maximum clone size is predicted to be 32, suggesting that larger patches we identified represent >1 independent clone whose descendants grew together.

### Delimited gene expression domains of the dorsal trunk, dorsal branch and transverse connective

To identify genes that are expressed in Tr2 branches during L3, we screened approximately 1300 enhancer trap lines whose expression was uncharacterized, as well as a group of candidate genes that express in domains of the embryo (*Chen et al., 1998*; *Franch-Marro and Casanova, 2002*; *Kuhnlein and Schuh, 1996*; *Llimargas, 1999*; *Thomas et al., 1991*) or larval (*Furriols and Bray, 2001*; *Pitsouli and Perrimon, 2010, 2013*) trachea for which either antibodies, enhancer trap lines,

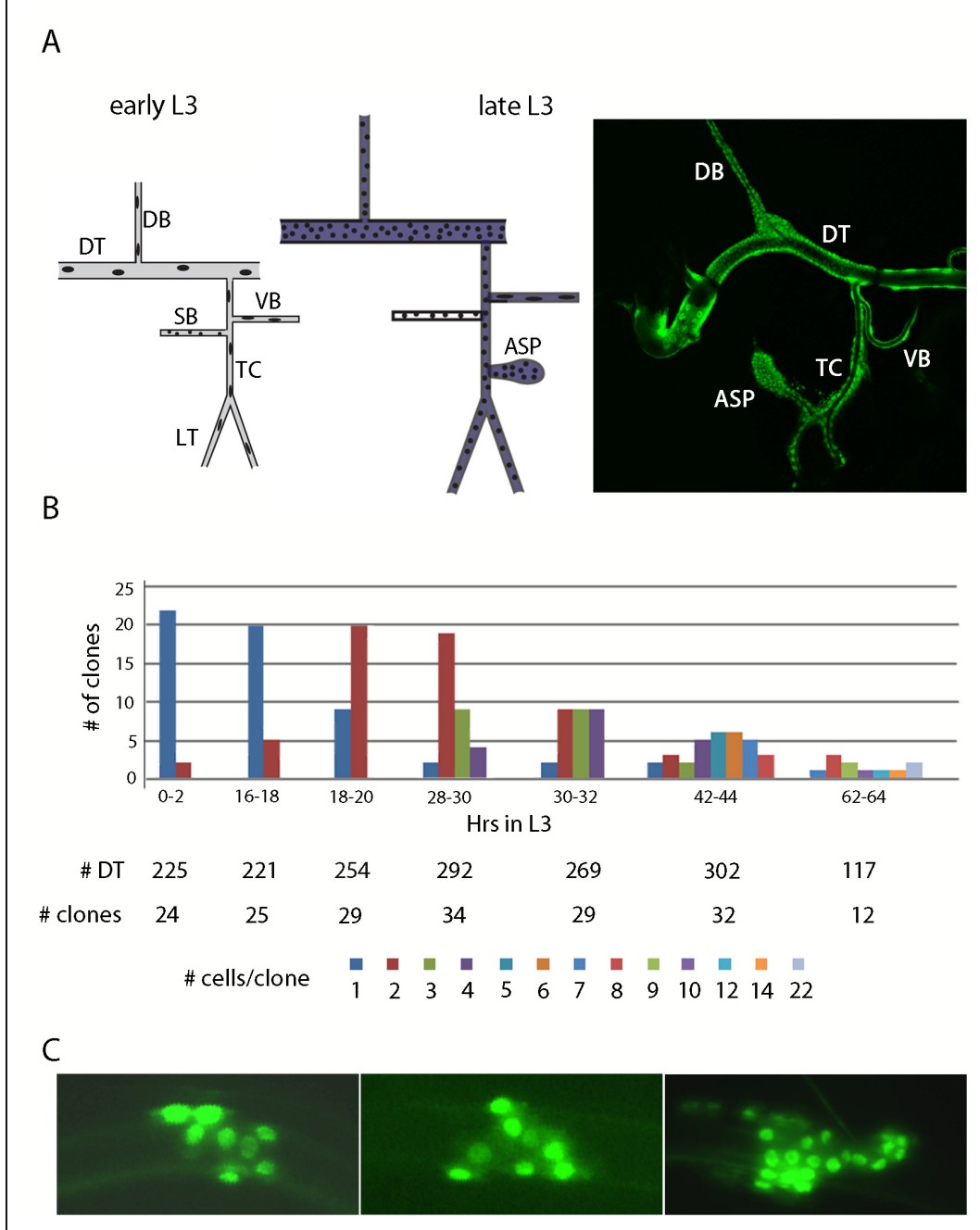

**Figure 1.** Cell divisions in the second tracheal metamere during the third instar. (**A**) Drawings of the Tr2 branches before (early L3) and after (late L3) the onset of cell divisions; dorsal trunk (DT), dorsal branch (DB), visceral branch (VB), spiracular branch (SB), transverse connective (TC), lateral trunk (LT), air sac primordium (ASP), region in which *btl-Gal4* is expressed (purple). (right panel) Nuclei in Tr2 visualized by fluorescence of GFP expressed by the *btl-Gal4* driver. (**B**) Bar graph representing clones of indicated sizes in the DT at the indicated times post L2-L3 molt; the numbers of DTs examined, numbers of cells in the clones and color code are listed below. (**C**) Three representative DT clones showing cell proliferation.

Gal4 lines, or enhancer reporter lines were available. *Figure 2* shows expression patterns for seven genes that are expressed in discrete regions of the L3 Tr2. These genes encode the transcription factors Sal, Cut and Kni, the Notch ligands Serrate (Ser) and Delta, the EGF ligand Vein, and Wingless (Wg). Sal, Ser and Delta expression was detected in DT cells, but not in cells of the DB (*Figure 2A–C*). There was a distinct expression limit of Ser and Delta-expressing DT cells near the TC junction (*Figure 2B,C*), but the limit of Sal expression in this region was not as distinct (*Figure 2A*). Ser was

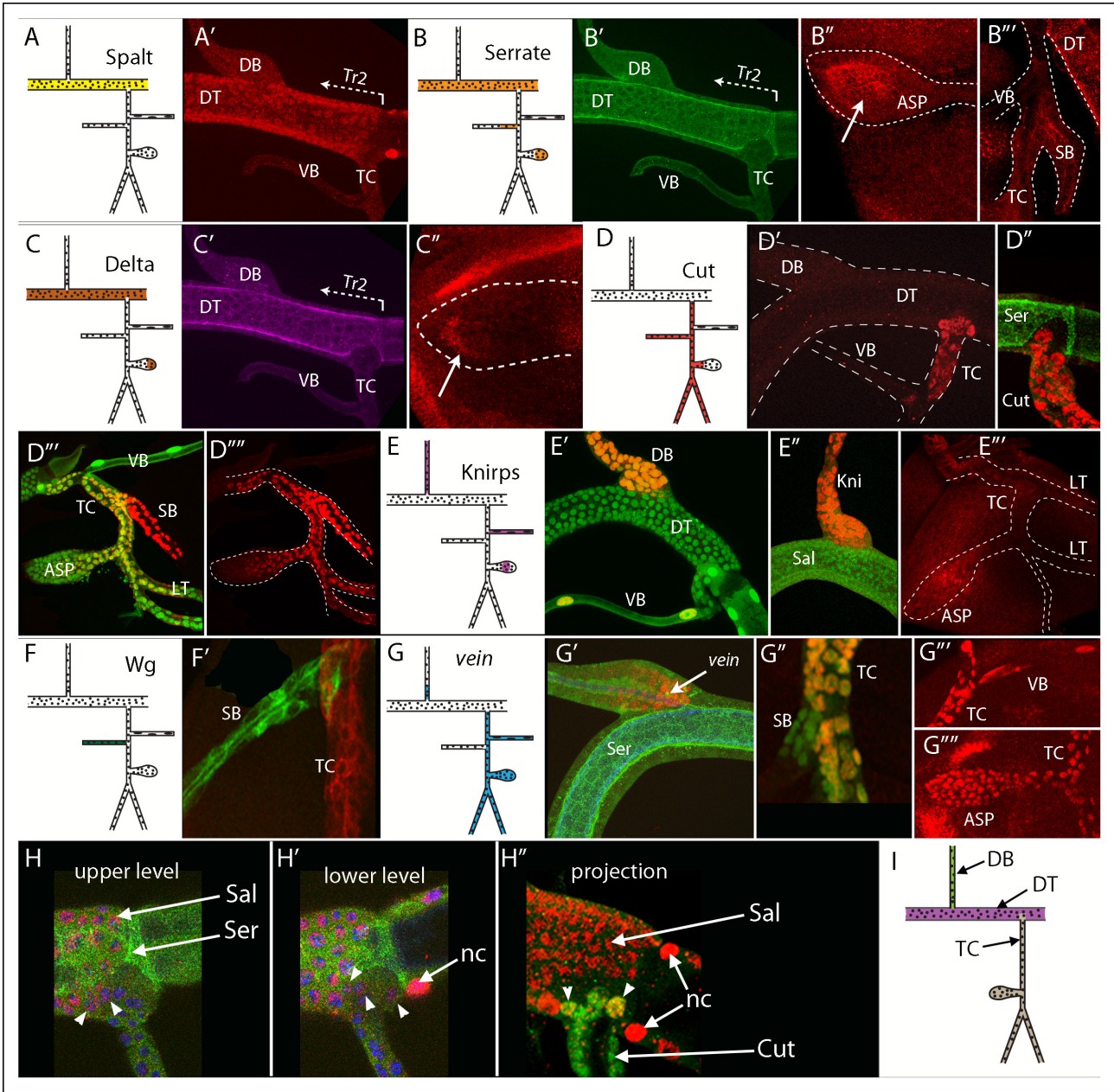

**Figure 2.** Patterns of gene expression in the second tracheal metamere of third instar larvae. (**A–C**) A Tr2 preparation stained for Sal, Ser and Delta; (**A**) Sal expression was specific to the DT; (**A'**) anti-Sal antibody stained the DT, but not DB, TC or VB; (**B**) Ser was expressed in the DT, ASP and SB; anti-Ser antibody stained the DT but not DB, TC or VB (**B'**), the distal ASP (**B"**) and the SB (**B"'**); (**C**) Delta was expressed in the DT and ASP; (**C'**) anti-Delta antibody stained the DT specifically and the distal ASP; (**D**) Cut was expressed in the TC, SB, ASP and LT; anti-Cut antibody stained the TC (**D'**) but not Ser-expressing DT cells (**D"**), and the SB, LT and proximal ASP (**D"'**, **D""**; Cut expression in the myoblasts was erased manually in the image file to highlight tracheal expression only); (**E**) Knirps was expressed in the DB, VB and ASP; anti-Knirps antibody stained nuclei in the DB and VB (**E'**) but not Sal-expressing DT cells (**E"**) and stained the medial region of the ASP (**E"'**); nuclei in (**D"'**), (**E'**) and (**G"**) contained GFP (*btl-Gal4 UAS-nlsGFP*); (**F**) *wg* was expressed in the SB; (**F'**) wg expression in the SB indicated by GFP fluorescence (*wg-Gal4 UAS-GFP*) but not in the TC (*btl-CD8:Cherry*); (**G**) vein was expressed in the proximal DB, TC, VB, ASP and LT; *vein* expression indicated by staining for *vein-lacZ* (red) in DB adjacent to Ser-expressing DT cells (**G'**), in the TC but not SB (**G"**), and in the VB, TC and ASP (**G"'**, **G""**). (**H, H'**) Sal expression (red, nuclear) and Ser expression (green, non-nuclear) detected in DT cells (arrows) in upper level and lower level optical sections; arrowheads indicate 5 cells in the TC domain that express Sal but not Ser; nc (node cell), DAPI-stained nuclei (blue); (**H"**) projection image showing Sal (red) and Cut (green) expressing TC cells; 2 cells in the TC domain (arrowheads) stained for both Sal and Cut. (**I**) Drawing with the DT, DB and TC expression domains indicated in purple, green and brown, respectively. The clonal analysis has not established whether the VB and SB are expression domains distinct from the TC.

detected in the medial ASP and in the spiracular branch (*Figure 2B*), and Delta expression was detected in the distal ASP (*Figure 2C*). We detected Cut expression throughout the spiracular branch, TC and the LT, and in the proximal ASP (*Figure 2D*), Kni expression in the DB, visceral branch and medial ASP (*Figure 2E*), Wg expression in the spiracular branch (*Figure 2F*) and Vein expression throughout the visceral branch, ASP, TC and LT and in the proximal DB (*Figure 2G*).

Several borders that define domains of expression for these genes appeared to coincide. For example, at the junction of the DT and DB, the DT cells, which expressed Sal, Ser and Delta (*Figure 2A',B',C'*) but not Kni, were precisely juxtaposed to Kni-expressing DB cells (*Figure 2E"*). Although the location of the border appeared to vary between different preparations, we attribute these varied appearances to the way the preparations rotated as they were mounted for viewing and to the fact that proximal/distal position of the border is not precisely the same around the circumference of the branch junction; in no specimen did we detect cells that expressed Kni together with Sal, Ser, or Delta.

Another common border appeared to separate the DT and TC. Cells on the trunk side of this border expressed Ser and Delta; cells on the other side expressed Cut and Vein. This border is more complex than the DT/DB border in several respects. First, the border extends a significant distance into the area of the DT and does not align with the DT/TC junction. Cells in this area of the DT are members of the TC lineage domain. Second, cells located in this part of the TC domain varied in number and size, and in contrast to the other Tr2 branches, there were large cells in this region in most wandering stage L3s. The number of large cells varied between 0–3. This may be a consequence of delayed entry into mitotic cycling by these cells and the possibility that some of the wandering L3s that we analyzed had not yet reached the stage when these cells start to divide; but we have not established the reason for the variability. Third, whereas all cells in the DT lineage domain expressed Sal and Ser at high levels and no cells in the TC lineage domain expressed either gene at high levels, in some samples, low level expression of Sal was detected in 1–2 TC domain cells that were adjacent to the DT domain on the "lower layer" next to the wing disc (*Figure 2H,H',H"*). All cells in the region of the TC domain in the DT expressed Cut and Vein. We note that there are also small regions of gene expression or lineage ambiguities at the anterior/posterior (A/P) compartment borders of the wing and eye-antennal imaginal discs (*Blair, 1992*; *Morata and Lawrence, 1979*), and suggest that the TC is also a lineage domain despite the apparent ambiguity for Sal at the DT/TC boundary. The TC lineage domain is defined by expression of Cut and Vein and by lack of expression of Ser and Delta, and it includes a portion that is physically within the DT. Based on this model, we suggest that Tr2 has distinct domains of gene expression that correlate with the DB, DT and TC. No distinct domains of gene expression that delimited the ASP from the TC were observed. *Figure 2I* depicts these DT, DB and TC domains.

## Notch activation at branch junctions

To characterize the role of Ser and Delta in the L3 Tr2, we monitored Notch signal transduction with the NRE-*lacZ* Notch pathway transcriptional reporter (*Figure 3A*) (*Furriols and Bray, 2001*). Previous reports describe Notch signaling and Notch reporter expression in branching morphogenesis and in specifying the number of fusion cells during embryo tracheal development (*Ghabrial and Krasnow, 2006*; *Ikeya and Hayashi, 1999*; *Llimargas, 1999*; *Steneberg et al., 1999*), and Notch signaling has been described to be generally present at tracheal branch junctions of L3 trachea (*Furriols and Bray, 2001*) and has been characterized in the spiracular branches (*Pitsouli and Perrimon, 2013*). Studies of Notch signaling in the Tr2 metamere have not been reported. We examined stages of embryo development subsequent to fusion of the dorsal trunk (post stage 16), and detected expression of NRE-*lacZ* at both the DT/DB and DT/TC junctions (*Figure 3B*). In the L3 Tr2, NRE-*lacZ* expression was also detected in the DT/DB and DT/TC junctions, as well as in the ASP, in the TC adjacent to the spiracular (*Figure 3C, D*) and in 1–2 cells of the visceral branch proximal to the TC (not shown). Notch signaling in the ASP is activated by Delta that is expressed in ASP-associated myoblasts (*Huang and Kornberg, 2015*); we did not investigate the function of Delta or Serrate expression by ASP cells or the source of the activating ligand for Notch activation in the TC or visceral branch.

At the DT/DB and DT/TC junctions, NRE-*lacZ* expression coincided precisely with the boundaries that are defined by the expression domains of Cut, Kni, Delta and Sal (*Figure 3E–H*). All of the NRE-lacZ expressing dorsal branch cells expressed Kni; all of the NRE-*lacZ* expressing TC cells expressed

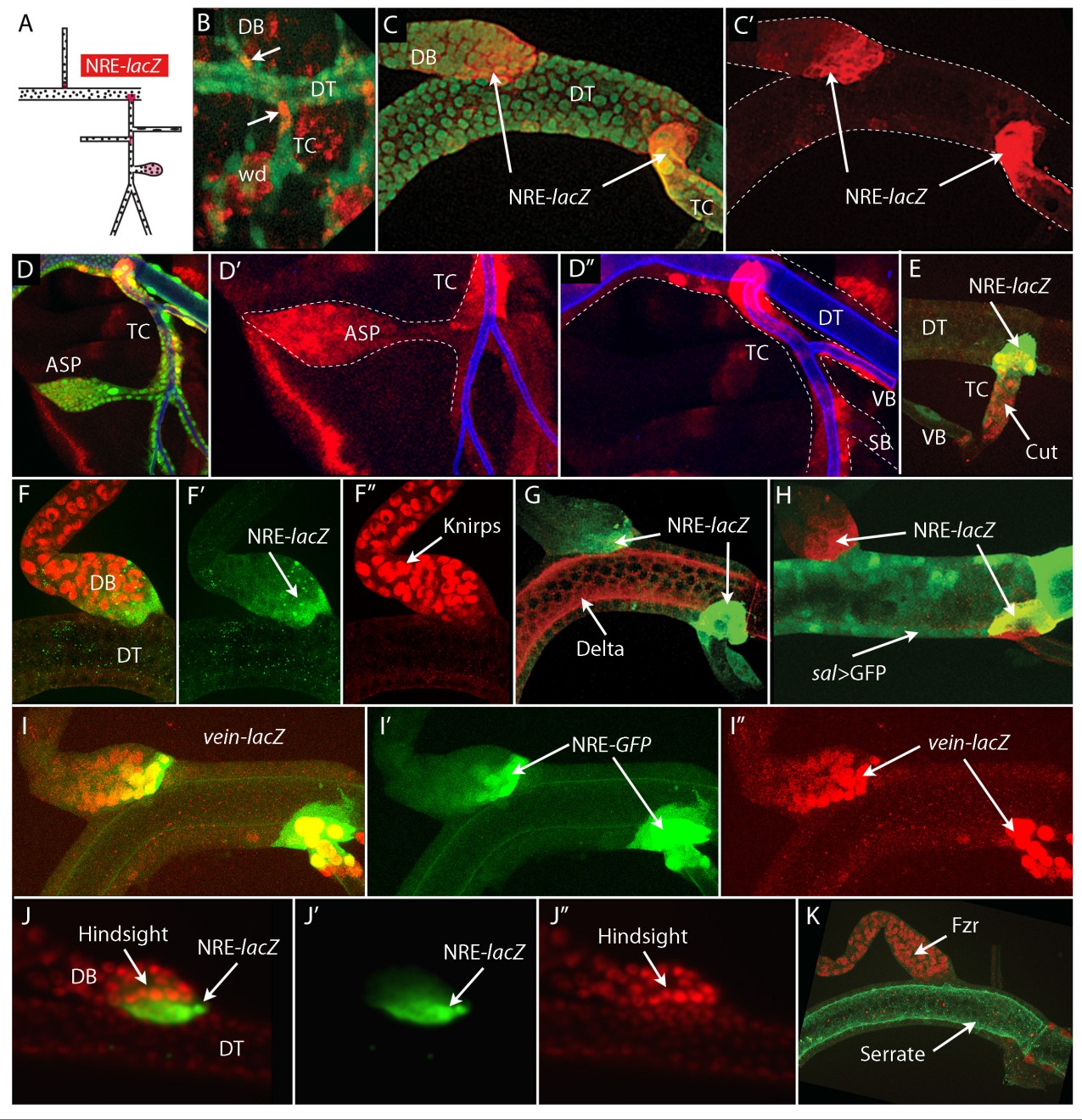

**Figure 3.** Discrete regions of Notch activation in the second tracheal metamere. (**A**) Drawing of a late L3 Tr2 metamere with red areas indicating the regions that express the Notch reporter NRE-*lacZ*; (**B**) Tr2 of a stage 14–16 embryo with tracheal nuclei marked with GFP fluorescence (*btl-Gal4 UAS GFP*), stained with anti-β-galactosidase antibody to identify cells expressing NRE-*lacZ* at DT/DB and DT/TC junctions (arrows); (**C, C'**) NRE-*lacZ* expression (red) at DT/DB and DT/TC junctions of L3 Tr2; Low (**D**) and high magnification (**D' D''**) images showing NRE-*lacZ* expression in the ASP, TC and VB; (**E**) Double stained preparation shows coincidence of NRE-*lacZ* and Cut expression in the TC; (**F-–F''**) images showing coincidence of NRE-*lacZ* and Knirps expression in the DB; (**G**) image showing juxtaposition of NRE-*lacZ* expression in the DB and TC with Delta; (**H**) image showing *sal* expression juxtaposed with NRE-*lacZ* expression in the DB and overlapping with NRE-*lacZ* expression in the TC; (**I–I''**) images showing coincidence of NRE-*GFP* and *vein-lacZ* expression at the DT/DB and DT/TC junctions; (**J–J''**) images showing juxtaposition of DB cells that express high levels of NRE-*lacZ* with cells that express the Notch target Hindsight; (**K**) the Notch target Fzr is expressed in DB cells adjacent to cells that express high levels of NRE-*lacZ*.

Cut. These results show that boundaries that define the DB, DT and TC gene expression domains are sites of Notch signaling. In the DB, expression of NRE-*lacZ* was highest in the cells that abut the Ser/Delta/Spalt expressing DT cells, and it decreased with increasing distance from the boundary. Similarly, expression of NRE-*lacZ* in the "TC domain" was highest in the cells that are in the DT, and expression decreased with increasing distance from the boundary. Expression of *vein* and of the Notch targets *Hindsight* and *Fizzy-related* appeared to correlate with the level of Notch activation in the proximal DB (*Figure 3I–K*). *vein* expression was highest in the cells with the most NRE-*GFP* expression, but expression of *Hindsight* and *Fizzy-related* in the DB was not detected in the cells with highest levels of Notch activation. These results suggest that Notch signaling may pattern the proximal DB.

### Lines of lineage restriction at the boundaries of the dorsal branch, dorsal trunk and transverse connective gene expression domains

We analyzed cell growth behavior in the DT, DB and TC by inducing marked clones and mapping their distribution. Similar clonal analysis studies of the wing imaginal disc revealed that in different discs, clones occupied varied locations and produced varied shapes in the wing, indicating that the descendants of particular single cells do not generate designated areas (*Bryant, 1970*; *Garcia-Bellido et al., 1973*). The clone borders were "wiggly" except at compartment borders where they were straight (reviewed in *Lawrence and Struhl, 1996*). Although the tracheal branches are tubes, not epithelial sheets, we were able to map clones in the DT, DB and TC. Most of the clones arose in the DT (as expected because of the greater relative number of founder cells), and the number of cells around its circumference was large enough that we were able to evaluate the contours of DT clones.

We generated marked clones using eight different regimens that varied clone type (e.g., "standard flipout", dual flipout clones, MARCM, and *M* MARCM) as well as time and length of induction (*Table 1*). The clone frequency for regimens A and F were low and few specimens had more than one marked patch. Clone frequency for regimens B-E, G, H was greater: most DTs had clones, most

**Table 1.**

| Regimen | Marking system | # Specimens | # Marked DT | HS stage | HS | DT/DB border clones | | | DT/TC border clones | |
|---------|----------------|-------------|-------------|----------|-----|---------|---------|-------|---------|---------|
| | | | | | | DT side | DB side | DT DB | DT side | TC side |
| A | GFP, NRE lacz | 106 | 40 | 48-50h AEL | 8' 35° | 3 | 0 | | 5 | 0 |
| B | GFP, NRE-lacz | 59 | 24 | 24-32h AEL | 5' 37° | 7 | 1 | | 8 | 0 |
| C | GFP | 29 | 11 | 24-32h AEL | 5' 37° | 2 | 1 | | 1 | 1 |
| D | MARCM, α-Delta | 52 | 43 | 4-6h AEL | 60' 38° | 1 | 3 | | 4 | 5 |
| E | MARCM, α-Cut | 48 | 34 | 4-6h AEL | 60' 38° | 6 | 2 | | 6 | 5 |
| F | M MARCM, α-Ser | 251 | 30 | 4-6h AEL | 30' 38° | 3 | 2 | | 2 | 3 |
| G | GFP & LacZ, α-Ser | 119 | 114 (GFP) 79 (LacZ)* | 24-26h AEL | 15' 37° | 11 | 11 | 6** | 8 | 0 |
| H | GFP & LacZ | 116 | 61 (GFP) 34 (LacZ)* | 24-26h AEL | 6' 37° | 9 | 5 | 0 | 6 | 1 |
| TOTAL | | 780 | 361 | | | 42 | 25 | 6 | 40 | 15 |

Genotype

A: NRElacz/hsFLP; actin>y >GAL4,UAS-GFP/ ; /MKRS

B: NRElacz/hsFLP; actin>y >GAL4,UAS-GFP/ ; /MKRS

C: hsFLP/Y; actin>y >GAL4,UAS-GFP/ ; /MKRS

D: hsFLP122,tubGAL4,UAS-NLS-GFP/ ; tubGal80 FRT40a/FRT40a; /

E: hsFLP122,tubGAL4,UAS-NLS-GFP/ ;tubGal80,FRT40a /FRT40a; /

F: hsFLP,tubGAL4, UAS-GFP-NLS; / ;RpS17,tubGal80,FRT80a/FRT80a

G: hsFLP/ orY; actin>y >GAL4,UAS-GFP/ ;actin>stop>lacZ-NLS/MKRS

H: hsFLP/ orY; actin>y >GAL4,UAS-GFP/ ;actin>stop>lacZ-NLS/MKRS

\* # DTs with clones at or near the DB and TC borders

\*\* specimens with independently marked DT and DB clones

DTs with clones had an average of more than one discrete patch of marked cells, and because many marked areas included more than 32 cells (the period during L3 when the cells divide after clone induction limits the number of mitoses to a maximum of five), these areas must be the descendants of more than one founder cell and represent more than one clone. Nevertheless, the clonal patches we observed in the DT, DB and TC appeared to have located randomly in these branches, and clones in the DT appeared to have "wiggly" borders except at the boundaries that delimit the DT, DB and TC expression domains. Among the 361 specimens we analyzed that had marked cells, 202 marked patches were identified that lined a portion of these borders. Ten example specimens are shown in *Figure 4* and the others are in the *Figure 4—figure supplement 1–7*. The varied size and location of the clones in *Figure 4* is evident in these examples that were selected to show representative clones that abut the DT/DB (*Figure 4A–F*) and DT/TC (*Figure 4G–J*) borders. These borders were identified by morphology, by expression of NRE-*lacZ* (*Figure 4B,G*), and in *Figure 4 (C,D)*, by the juxtaposition of two clones that had been independently generated and differentially marked, and precisely correlated with Ser expression.

Of the 79 marked patches that lined the DT/DB border from one side or other, all had many cells (*Figure 4—figure supplement 1–3*). Although 26 other specimens had marked cells on both sides of the DT/DB border (*Figure 4—figure supplement 7*), these specimens had multiple large patches of marked cells that in combination averaged 34% ( ± 23%, std. dev.) coverage of the DT surface. It is most likely therefore that the patches that meet on either side of the DT/TC border represent more than one clone and therefore are not inconsistent with the model we propose - that the DT/DB border is also a line of clonal restriction.

87 marked patches were identified that lined the DT/TC border from either side (*Figure 4—figure supplement 4–6*). All 40 DT and 15 of the TC clonal patches had multiple cells, but as noted above, most specimens had 1–3 large cells in the TC domain within the DT, and 1–2 of these were marked in 32 of the 87. 15 specimens were identified that had marked cells on both sides of the DT/TC border (*Figure 4—figure supplement 7*); in 13 of these, the only marked TC cell was a single large one. Because the number of specimens with marked DT patches at the border (55) was similar to the number of specimens with marked TC cells (62), we conclude that the number of cells on either side of the DT/TC border at the time of recombination was approximately the same. And because the number of specimens with marked cells on both sides of the DT/TC border (15/102) relative to those with clones only on one side (DT only (40) TC only (47) = 87/102) is consistent with the expected number of independent clones in the DT and TC domains, we suggest that the DT/TC border is also a line of clonal restriction.

## Spalt function, branch identity and boundary formation

To investigate which functions might be served by the genes that are expressed specifically in the DT, DB and TC domains, we analyzed clones that either lacked sal function or ectopically expressed *sal*. In contrast to marked control clones in the DT, which integrated with their unmarked neighbors and were not morphologically distinct (*Figure 5A*), clones that lacked sal function appeared to sort out from their neighbors and to bulge abnormally from the plane of the trunk tube (*Figure 5B–F*). The *sal* mutant cells expressed Kni and *vein* (*Figure 5C,D*) but did not express either Delta or Cut (*Figure 5B,E*). All the mutant cells expressed NRE-*lacZ* (*Figure 5F*). We also analyzed clones that ectopically expressed *kni*, and they also appeared to sort out; and they expressed *vein* (*Figure 5G*). This expression signature (*vein*, Kni, Sal⁻, Delta⁻, Cut⁻) is the same as that of DB cells, suggesting that sal function is required for DT identity and that without sal function or with Kni, they transform to DB identity. The sorting out phenotype is consistent with the idea that the presence of mutant cells created an ectopic juxtaposition of cells with different identities. Expression of NRE-*lacZ* in the mutant cells that lacked sal function (*Figure 5F*) suggests that Notch signaling was activated at the ectopic borders that formed where the two groups of cells abut, just as it does at the normal DT/DB junction.

We also analyzed clones that ectopically expressed Sal, and noted three phenotypes associated with clones in the DB. First, cells adjacent to Sal-expressing cells strongly expressed *vein* (*Figure 5H*), which is normally expressed at high levels in the DB only by cells near the DB/DT junction that activate Notch signaling (*Figure 5C,F*). Second, Sal-expressing cells expressed Delta (*Figure 5I*), which is normally expressed in Sal-expressing DT cells and not in the DB (*Figure 2A,C*). Third, the diameter of the branch in the region affected by Sal-expressing clones was reduced in the

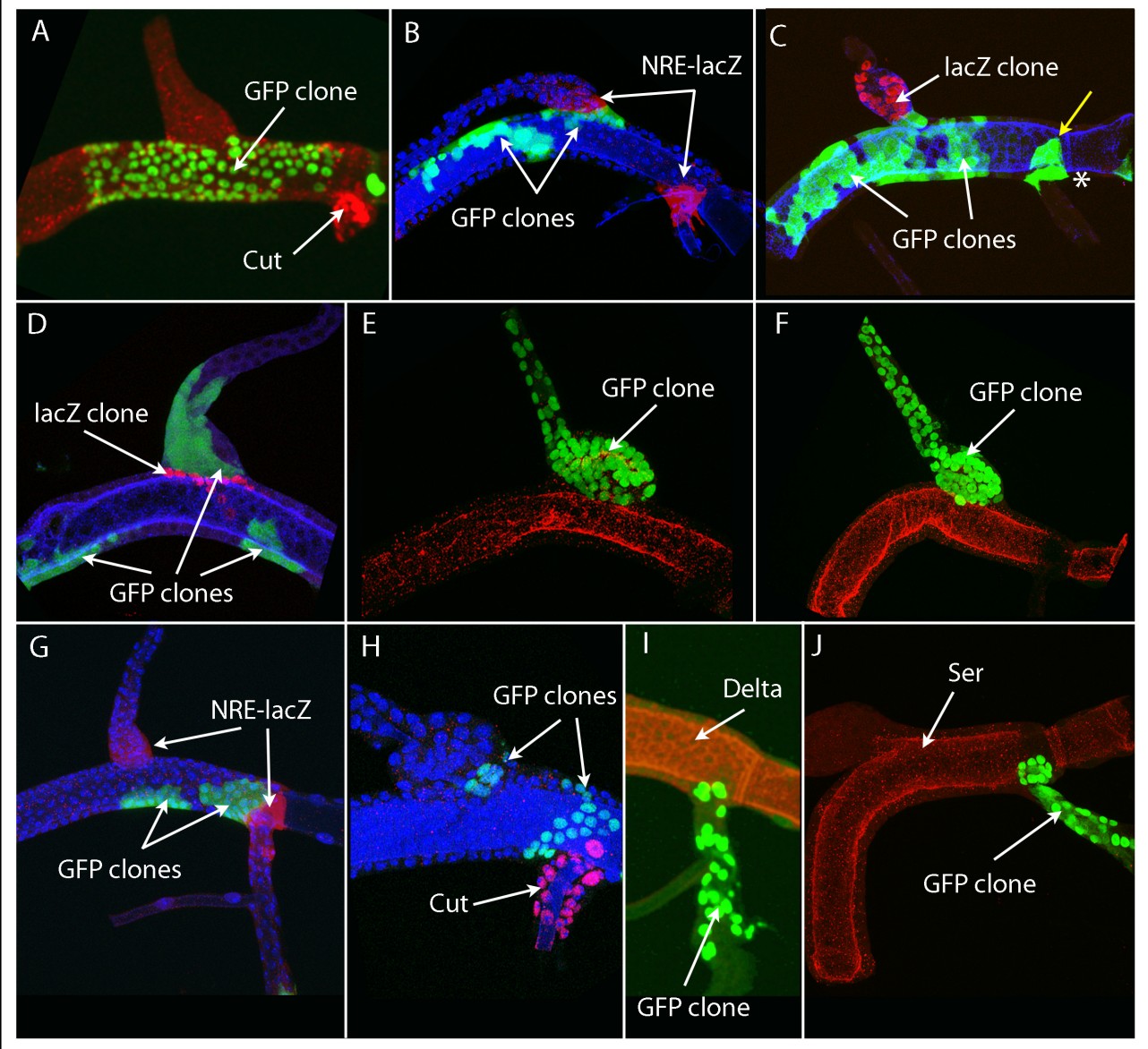

**Figure 4.** Marked clones that meet but do not cross into or from the dorsal trunk. (A-–D) Marked clones in the DT line the DT/DB junction. (A) large patch of GFP-expressing cells abuts the DT/DB junction; Cut expression in TC (red); clone induction by regimen Experiment E (*Table 1*). (B) several GFP expressing clones in DT (Expt. A), one of which abuts the DT/DB junction defined by NRE-*lacZ* expression (red). (C) several GFP expressing clones in DT (white arrows), one of which abuts the DT/DB junction and is juxtaposed to a LacZ-expressing DB clone; a single cell clone expressing GFP in the TC (yellow arrow) abuts the DT/TC junction and a GFP-expressing TC clone (*); Ser expression (blue); (Expt. G). (D) LacZ-expressing DT clone abuts the DT/DB junction and is juxtaposed to a GFP-expressing DB clone; Ser expression (blue); (Expt G). (E, F) GFP expressing DB clones that abut the DT/DB junction (Expt. F). (G–J) GFP expressing clones that abut the DT/TC junction that was also defined by expression of NRE-*lacZ* (G; Expt. B), Cut (H; Expt. E), Delta (I; Expt. D) and Ser (J; Expt. F). (B, G, H) DAPI (blue).

The following figure supplements are available for figure 4:

**Figure supplement 1.** Dorsal trunk clones at the DT/DB border.
**Figure supplement 2.** Dorsal branch clones at the DT/DB border.
**Figure supplement 3.** Dorsal trunk and dorsal branch clones abutting the same DT/DB border.
**Figure supplement 4.** Dorsal trunk clones at the DT/TC border.

*Figure 4. continued on next page*

*Figure 4. Continued*

**Figure supplement 5.** Transverse connective clones abutting the DT/TC border.
DOI: 10.7554/eLife.08666.012
**Figure supplement 6.** Large marked cells in the TC domain of the DT.
DOI: 10.7554/eLife.08666.013
**Figure supplement 7.** Marked patches of cells that straddle the DT/DB and DT/TC borders.
DOI: 10.7554/eLife.08666.014

cells that expressed Sal and greater in adjacent regions that expressed *vein* (*Figure 5H*). This morphology has features in common with the normal DT/DB junction that has an expanded diameter at the proximal DB, *vein*-expressing side. Clones in the TC that over-expressed *sal* also up-regulated Delta (*Figure 5J*) and down-regulated Cut (*Figure 5K*). The effects of ectopic *sal* expression in the DB and TC suggest that *sal* expression transformed their cells to a DT identity.

### Notch signaling in the dorsal branch

The role of Notch at the DT/DB junction was characterized further by examining the role of Delta expression in the DT and of Notch signaling in the DB. Clones that ectopically expressed Delta in the DB non-autonomously activated *vein-lacZ* expression and Notch signaling in adjacent cells, and the affected regions had a diameter greater than the normal DB (*Figure 6A,C*). *vein* was also expressed in DB clones that ectopically expressed Notch[ACT], and the morphology of the region with the mutant cells was abnormal and more characteristic of the expanded proximal DB (*Figure 6B*). In contrast, Delta-expressing clones in the DT, which also activated Notch signaling in adjacent cells, did not alter the morphology of the DT (*Figure 6D*). Because *vein* is normally expressed by DB cells in the expanded region at the DT/DB junction, these phenotypes suggest that ectopic over-expression of Delta in the DB transformed adjacent cells to a junction identity and that normally, Delta-activated Notch signaling at the DT/DB junction induces *vein* expression and morphological specialization. This conclusion is consistent with the effects of Delta over-expression on another morphological feature of the expanded region of the proximal DB. Anti-cadherin staining revealed that the lumen in this region of the wild type DB is branched (*Figure 6E*). In addition to the lumen that extends the length of the DB, a short segment of lumen angles obliquely in the direction of the DT. A similar structure was detected in the expanded region associated with Delta-expressing clones (*Figure 6F*). It was present in the cells adjacent to the clone, and its orientation was toward the clone (mirror image with opposite polarity relative to the normal branch).

We expressed Notch RNAi in trachea (*btl-Gal4 UAS-NotchRNAi*) and observed that the presence of Notch RNAi had no apparent effect on Delta expression in the DT, but it reduced NRE-*lacZ* expression to undetectable levels at both the DT/DB and DT/TC junctions, and altered the morphology of the DB (*Figure 6G*). In the absence of Notch signaling, the proximal DB was not expanded and had the same diameter as other more distal regions of the DB. In the *NotchRNAi*-expressing trachea of wandering L3 larvae, the number and density of cells in the Tr2 branches appeared to be less than normal, and may reflect developmental delay. Comparison of trachea in younger, control L3 larvae (40–42 hrs post L2-L3 molt) that had a similar density of cells in the DT revealed that the proximal DB was normally broadened at this earlier stage (*Figure 6H*). We conclude that Notch signaling is required at the DB junction for normal morphogenesis.

To investigate whether Notch signaling is necessary for lineage segregation at the compartment border, clones were induced that express *kni, NotchRNAi* and GFP. *Figure 6I* shows a DT clone in a preparation that was stained with antibodies against Kni and Sal (which marks nuclei of DT cells). The nuclei in the clone lacked Sal staining (similar to DB cells), and in contrast to clones that ectopically over-express *kni* in an otherwise normal genetic background (*Figure 5G*), the clones that also down-regulate Notch do not sort out, but appear to integrate with their neighbors without forming a bulge. The clone also crossed into the DB.

### Discussion

The Tr2 metamere is transformed by programs of cell division during the L3 period (*Guha and Kornberg, 2005*; *Guha et al., 2008*). It is the only Drosophila organ generated by a tubulogenesis

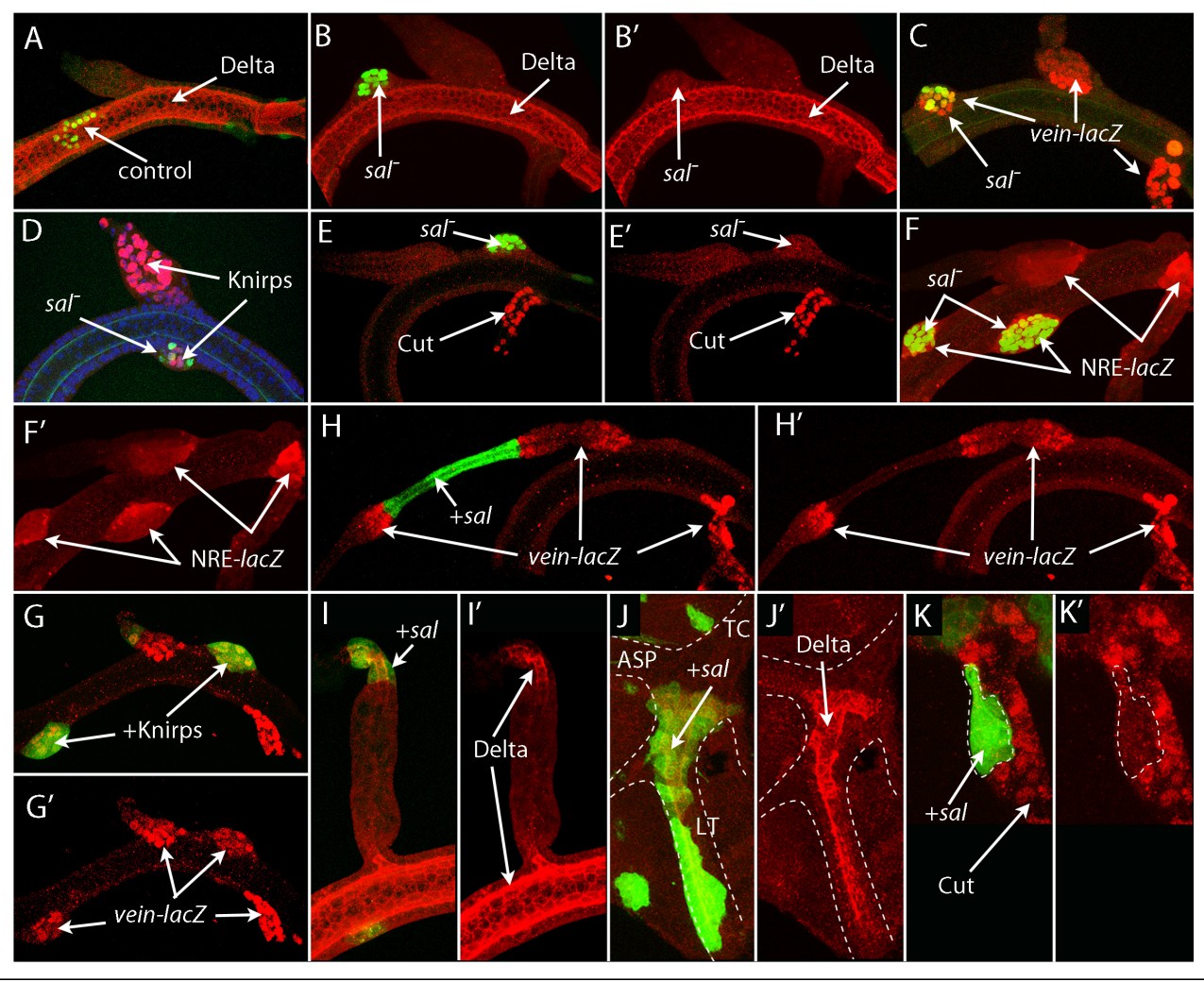

**Figure 5.** Mutant clones transform dorsal branch, dorsal trunk and transverse connective cells. (A) Control DT clone of GFP-expressing cells with no effect on morphology or expression of Delta; (B—F') Clones of *sal* mutant DT cells generated bulges, apparently sorting out, and lacked Delta expression (B, B'), ectopically expressed *vein-lacZ* (C) and Knirps (D), not normally expressed by DT cells, but not Cut (E, E') and activated the NRE-*lacZ* reporter (F, F'). (G, G') Clone ectopically expressing Knirps in the DT activated *vein-lacZ* expression. (H, I) Clones that ectopically expressed Sal in the DB reduced the diameter of the DB and induced neighboring wild type cells to express *vein-lacZ* (H, H'') and ectopically expressed Delta (I, I''). (J, K) Clones that ectopically expressed Sal in the TC ectopically expressed Delta, which is not normally expressed in the TC (J, J'), but did not express Cut, which is normally expressed in the TC (K, K').

process that combines both cell division and morphogenesis, and it is therefore the only system to which the power of Drosophila developmental genetics can be applied to address the question that instigated the study described here – are cells that re-initiate cell cycling in a fully differentiated tubular organ (specifically in the L3 Tr2) descendants of un-differentiated subgroups (stem cells?), or do the differentiated cells re-enter mitotic cycling? Our initial assumption was that the tracheal remodeling is a product of pools of adult tracheoblasts that grow and assume a functional role only after the larval periods, but our previous studies showed that DT re-population in Tr2 involves the activation of mitotic cycling in differentiated larval cells (*Guha et al., 2008*; *Sato et al., 2008*). However, these studies did not establish whether the dividing cells represent a selected and specialized subgroup, or whether re-entry into the cell cycle was a general feature of the DT cells. The results of the clonal analysis reported here supports the idea that most or all cells in the early L3 DT contribute to re-population, and because their descendants can fully account for repopulation, we conclude that there is no other cell type involved. The varied locations and shapes of the clones suggest that

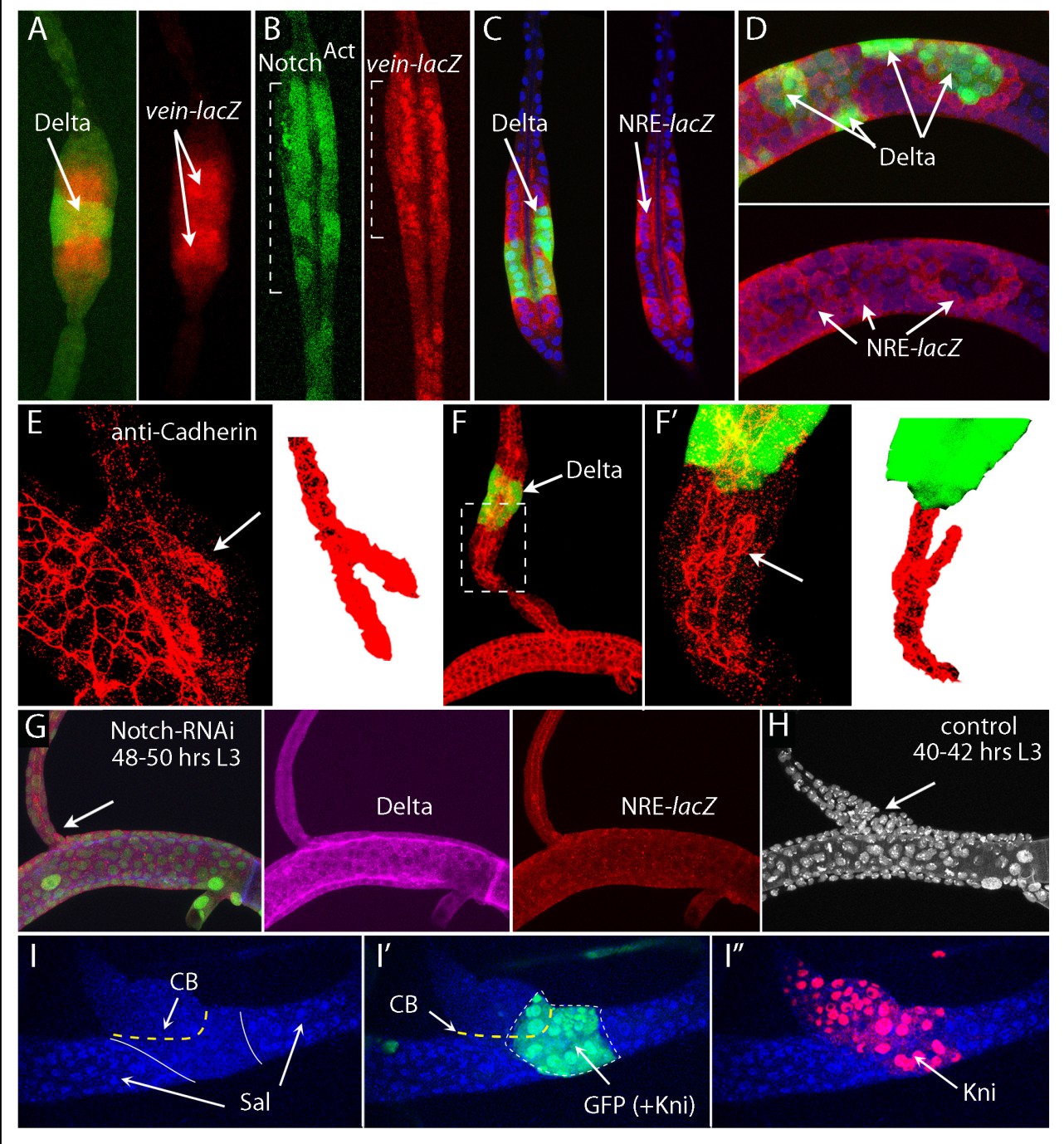

**Figure 6.** Notch signaling is required by dorsal branch cells at the junction with the dorsal trunk. (A) Clone of DB cells that ectopically expressed Delta (green) increased the diameter of the DB and induced wild type neighbors to express *vein-lacZ* (red), normally expressed in the DB only by cells at the DT junction. (B) Clone of DB cells (indicated by brackets) that ectopically expressed Notch[ACT] (green) increased the diameter of the DB and activated *vein-lacZ*. (C, D) Clones of DB (C) and DT (D) cells that ectopically expressed Delta (green) activated NRE-*lacZ* expression in adjacent cells. (E) Image of the proximal DB stained with anti-DE-Cadherin antibody; bifurcated lumen indicated by arrow (left panel) and isolated and colored red (right panel). (F) Delta expressing clone (green) in the DB induced a bifurcated luminal structure with orientation opposite to normal. (F') Higher magnification view of boxed area of (F). (G) Expression of Notch-RNAi did not affect DT morphology or Delta expression (magenta) but reduced NRE-lacZ expression at both DT/DB and DT/TC junctions and reduced the diameter of the proximal DB (arrow). Number of DT cells was reduced to levels characteristic of normal younger larvae whose proximal DB is expanded (H). (I-I") Clone that ectopically expressed *kni, Notch*-RNAi and GFP in the DT did not express Sal (I, blue) and did not sort out from DT cells (I, I"). DT/DB compartment border (CB) indicated by dashed yellow line, borders of Sal expression by solid white line (I) and border of clone by dashed white line (I').

the cells in the L3 DT constitute an equivalence group. Previous studies of re-population of the DB came to a similar conclusion (*Weaver and Krasnow, 2008*).

The clonal analysis yielded an unexpected property of Tr2: DT clones appeared to respect boundaries near the junctions of the DB and TC. We analyzed marked clones generated by eight regimens of recombination, including one that produced *M* clones in a *M* background and endows the marked cells with a growth advantage (*Morata and Ripoll, 1975*), and two that produced GFP- and LacZ-expressing clones independently in the same animal. 79 clones were identified that appeared to respect a common border at the DT/DB junction and 55 clones (with >1 marked cell) were identified that appeared to respect a common border at the DT/TC junction. A feature of these clones is the contrast between portions of their perimeter that appeared to be "straight" and abut the border, and the remaining portions of their perimeter, which were more irregular. This feature has been previously noted for the borders of developmental compartments (*Dahmann and Basler, 2000*; *Dominguez and de Celis, 1998*; *Lawrence, 1997*), and the question we consider is whether the apparent lineage borders at the DT/DB and DT/TC junctions (*Figure 2I*) represent borders of developmental compartments.

Developmental compartments have been characterized in the first abdominal segment of the adult, the wing, leg, eye-antennal and genital imaginal discs and the embryo. Their key distinguishing properties are that compartment borders confine all the descendants of their founding cells, and all their cells express a "selector" gene or set of genes that determine their identity (*Garcia-Bellido, 1975*). The compartment borders restrict the growth of their constituent cells to a defined geographical area and compartments are domains of gene expression.

We address the relevant features of the Tr2 branches by focusing on the DT. The Tr2 DT is contiguous with the DT metameres of Tr1 and Tr3 and is constituted by a single layer of epithelial cells. Doughnut-shaped node cells form the seams where the metameres join, but unlike the other cells of the DT, the Tr2 node cells do not divide during L3. Although we did not identify clones formed by dividing Tr2 DT cells that crossed into either Tr1 or Tr3, this cannot be considered evidence of lineage restriction because the non-dividing node cells may have, by their presence, barred expansion of clones into the adjacent metameres. In contrast, we detected no morphological feature that appeared to separate the DT from DB or TC cells into distinct populations, and the clonal analysis is not consistent with the presence of a group of non-dividing cells at the DT/DB and DT/TC junctions.

Although the non-planar nature of the intersection of the DT and DB complicates imaging because the rotational orientation was not the same for each specimen that was mounted for viewing, we found 79 clones that appeared to obey a common line of restriction at the DT/DB junction. Many of the marked patches were large (five were *M* and had >50 cells) and many extended around at least half the perimeter of the junction. Six specimens were recovered that had two independently marked clones that met at the DT/DB junction. We suggest that these clones are strong evidence for a compartment border at the DT/DB junction and we found several other aspects of the junction that are consistent with this idea.

First, the clone borders that met the common line of lineage restriction appeared to be straighter than elsewhere. The "smooth" borders of clones at compartment boundaries is a characteristic that has been noted at the boundaries of the A/P and dorsal/ventral (D/V) compartments of the wing disc (*Dominguez and de Celis, 1998*; *Garcia-Bellido et al., 1973*), and at the A/P compartment boundaries of the leg disc (*Lawrence et al., 1979*) and first abdominal segment (*Kornberg, 1981a*). Although the smooth borders have been attributed to local cell interactions involving differential cell affinities or mechanical tension (*Dahmann et al., 2011*), their basis is not understood. We have argued that this property is a consequence of signaling centers that form to one side and are not a property of the border itself (*Chuang and Kornberg, 2000*).

Second, the line of lineage restriction coincided with DB and DT gene expression domains. We defined these domains by expression of *kni* in DB cells and of *sal*, *Ser* and *Delta* in DT cells, and found that sal function is essential in DT cells for their identity and that *sal* expression in DB cells is sufficient to transform to a DT identity. DT cells that over-express *kni* ectopically appear to "sort out", consistent with their transformation to a DB identity. These analyses suggest that cells in the DT and DB domains may be regulated by genes with "selector" function (*Garcia-Bellido, 1975*), and establish the coincidence of lineage restriction and the expression domains at single cell resolution.

Third, DB cells at the DT/DB border expressed the Notch reporter NRE-*lacZ* and were juxtaposed to DT cells that expressed the Notch ligands Delta and Ser but did not express NRE-*lacZ*. NRE-*lacZ*

expression was highest in the DB cells at the border and decreased in a graded manner in DB cells farther away. These properties suggest that the DT/DB border situates a Notch signaling center at this position. Both gene expression and morphology appeared to be polarized relative to the DT/DB border – suggesting that this border may be a site of polarity reversal, which is also a defining feature of the A/P compartment border of the wing disc (*Chen and Struhl, 1996*; *Chuang and Kornberg, 2000*; *Garcia-Bellido and Santamaria, 1972*; *Lawrence and Morata, 1976*; *Lawrence et al., 2007*; *Tabata et al., 1995*).

Fourth, the behavior of mutant clones suggests that segregation of DT and DB cells depended on Notch. *vein* and *kni* are normally expressed by DB border cells and not by DT cells, and cells in the DT that ectopically over-expressed *kni* appeared to "sort out" from their DT neighbors and expressed *vein*. In contrast, *kni*-expressing cells in the DT that also lacked normal Notch function did not sort out. They appeared to integrate with their neighbors and did not respect the compartment border. Consistent with the idea that Notch is involved in maintaining the segregation of DT and DB cells, conditions that ectopically activated Notch appeared to induce ectopic borders. DB clones that over-expressed *sal* ectopically activated Notch signaling in adjacent cells and DT clones of *sal* mutant cells induced Notch signaling and also appeared to sort out from their neighbors. Notch signaling is also required for the wing disc D/V compartment border (*Blair et al., 1994*; *Cohen et al., 1992*; *Diaz-Benjumea and Cohen, 1993*; *Irvine and Wieschaus, 1994*; *Micchelli et al., 1997*; *Rauskolb et al., 1999*), although the mechanism may not be the same. In contrast to the Tr2 borders where Notch activation is polarized and specific to only one side, Notch signaling is active on both sides of the wing disc D/V border.

The presence of a compartment border at the DT/TC junction is supported by the clones that define it from either side and by its coincidence with a Notch signaling domain. However the variable cell composition at the junction and the presence of large, non-mitotic cells on the TC side in many wandering stage larvae complicates interpretation of the apparent lineage restriction. We do not yet understand the basis for the variability, but assume that it represents a transient state and that these large cells begin to divide later than the others. If true, they likely prevent dividing cells on the DT and TC side from mixing, but this does not argue against the existence of a compartment border. The activation of Notch signaling and the fact that the Cut identity of the TC cells is sensitive to the presence of Sal support the idea that there is one and a model in which the DT represents a development compartment that is segregated from the DB and TC. Studies of the cell divisions that follow during pupal development may provide additional evidence for the lineage restriction we identified.

## Perspectives

The discovery of developmental compartments in the Drosophila wing imaginal disc was a major advance that provided a novel conceptual basis to understand how groups of cells are programmed and how genetic programs regulate development independently of overt morphology (*Crick and Lawrence, 1975*; *Garcia-Bellido et al., 1973*). Although the concepts were instrumental to understanding the function of the genes that segment the early embryo when these genes were later discovered, and although discoveries of developmental compartments in other imaginal discs and in the abdominal histoblast nests followed, their generality has remained an open question because the types of tissues in which they were found represent a limited subset. Studies that map gene expression in developing tissues have identified many examples in which expression is restricted to well-defined domains, and some studies have correlated these domains with cell lineage (*Buchon et al., 2013*; *Marianes and Spradling, 2013*), but it is not known whether these cases satisfy other criteria that have been ascribed to developmental compartments - such as association with signaling centers and developmental polarity. Although we do not know if all these criteria are important distinctions, we prefer to reserve the term developmental compartments for contexts that have them. Using this strict definition, the studies reported here are the first to identify developmental compartments in an internal tubular organ. We did not initiate these studies expecting to find developmental compartments, but discovered them in the course of a thorough clonal analysis. It is possible that similar studies may also find them in other internal organs, and in particular in the vascular system of vertebrates that are also branched and have regions with distinct identities.

# Materials and methods

## Drosophila stocks

This work unless otherwise indicated.

*btl-Gal4 UAS-nlsGFP* (**Guha and Kornberg, 2005**)

*NRE-lacZ* (**Furriols and Bray, 2001**)

*sal-Gal4 UAS-GFP* (**Makhijani et al., 2011**)

*yw hsflp;actin>CD2>Gal4;UAS-nlsGFP* (**Guha et al., 2008**)

*yw hsflp;actin>y+>Gal4 UAS-GFP;MKRS/TM6B*

*hsFLP tubGAL4 UAS-nlsGFP;+/+;M(3)i⁵⁵ tubGal80 FRT80B/TM6B* (Bloomington #42732)

*hsFLP tubGal4 UAS-nlsGFP;tubGal80 FRT40A/FRT40A tubGal80* (Bloomington #1816)

*FRT40A Df(2L)32FP-5/CyO (Df(2L)32FP-5 uncovers spalt and spalt-related)* (**Organista and De Celis, 2013**)

*UAS-Sal;UAS-CD8GFP* (Bloomington #29715)

*UAS-Delta* (Bloomington #26694)

*UAS-knirps* (**Chen et al., 1998**)

*UAS-N-RNAi* (NIG-Fly #3936-R2))

*UAS-Nact* (**Hwang and Rulifson, 2011**)

vein-lacZ (Bloomington #11749)

*yw hsflp;actin>y+>Gal4 UAS-GFP;NRE-lacZ*

*yw hsflp;actin>y+>Gal4UAS-GFP;vein-lacZ*

*actin>stop>nlslacZ* (Bloomington #6355)

*en GAL4 UAS-myr-mRFP, NRE-EGFP* (Bloomington #30729)

*fzr-lacZ* (Bloomington #12241)

*wg-Gal4 UAS-CD8:GFP/CyO* (**Huang and Kornberg, 2015**)

*btl-LHG/CyO;lexO Cherry-CAAX/TM6B* (**Roy et al., 2014**)

The enhancer trap lines (collection generated by the U. Heberlein lab) have insertions of the pGawB containing a Gal4 enhancer trap construct.

## Clonal analysis

For analysis of cell proliferation in the DT, FLP recombinase was induced by heat shock at 37° for 15 minutes during late embryonic stages, and animals were picked at the L2-L3 molt (55 hours later) and aged for defined periods before being sacrificed for analysis.

Flpout clones were obtained by heat shocking L1 larvae (24-32h AEL) at 37° for 5 minutes or L2 larvae (48-50h AEL) at 35° for 8 minutes. MARCM clones were induced in embryos with a 60 minute heat shock at 38°. MARCM clones in a *Minute* background were induced in embryos with a 30 minute heat shock at 38°. Dual clones (that either express GFP or LacZ-NLS) were induced by heat shocking L1 larvae (24-26hAEL) at 37° for 6 or 15 minutes. Because the frequency of clones induced by these regimens was high and most specimens had multiple clones, statistical measures could not be used to calculate the probability that marked patches of cells represent the descendants of one or several founder cells. The Supplements to *Figure 4* contain images of all specimens with clones at the DT/DB or DT/TC borders. Supplement 6 shows the 32 that have either one or two marked cells in the TC domain of the DT. Supplement 7 shows the 26 that have marked cells on both sides of the DT/DB border and the 15 that have marked cells on both sides of the DT/TC border (13 of which have marked cells in the TC domain limited to only one large cell). Clones ectopically expressing kni and Notch-RNAi were induced heat shock at 37° for 10 minutes (genotype: *yw hsflp/+;act5C>y +>Gal4 UAS-GFP/UAS-kni;UAS-Notch-RNAi/+*).

Loss-of-function MARCM clones of *salm, salr* mutant cells were generated by subjecting L2 larvae (48-50h AEL) to heat shock at 38° for 1hr; wandering L3 larvae were dissected for analysis.

Ectopic expression clones that expressed *salm, Delta, knirps, knirps* and *Notch*-RNAi, or *Notch^Act* were induced in 2-4 day old animals by heat shock at 37° for 10 minutes; wandering L3 larvae were dissected for analysis after 48 hours.

## Immunohistochemistry

Larvae were fixed in 4% PF followed by washing with 1x PBS with $Ca^{++}/Mg^{++}$. These larvae were blocked in blocking buffer (1xPBS, 0.5% Donkey /Goat serum and 0.1% Triton X for one hour). Primary antibody staining was performed using above-mentioned buffer for ~8 hrs at RT or ~ 16h at 4 degrees followed by three washes of 15 minutes in blocking buffer at RT. Secondary antibody staining was done in the blocking buffer for 1-2 hours and washed three times with blocking buffer for 15 minutes followed by DAPI staining for 30 minutes and 2 additional washes in blocking buffer. Thereafter samples were stored in 1x PBS at 4 degrees prior to dissection and samples of larval Tr2 trachea or wing discs were mounted in Vectashield. Primary

antibodies: mouse anti-Delta (C594.9B, 1:200), mouse anti-β-galactosidase (40A1, 1:100), mouse anti-Cut (2B10, 1:100), guinea pig anti-Knirps (1:200, gift from J. Reinitz), rabbit anti-Sal (1:50, gift from R. Schüh), rabbit anti-β-galactosidase (1:1000), rat anti-Serrate (1:1000), rat anti-Cadherin (DCAD2, 1:20). Secondary antibodies, at (1:500) or (1:1000): anti-mouse Alexa 488, anti-mouse Alexa 555, anti-mouse Alexa 647, anti-Rat Alexa 555, anti-Guinea pig Alexa 555, anti-Rabbit Alexa 555 anti-Rabbit Alexa488.

## Imaging and image processing

Leica SPE confocal was used to image the slides. 20x or 40x oil immersion objectives were used. Z-projections of images were compiled with Image J and Adobe Photoshop was used to merge channels. The area covered by the marked patches in the DTs of the 26 projection images in upper panel of *Figure 4—figure supplement 7*, were measured using ImageJ.

## Acknowledgements

We thank Z Ali-Murthy, M Calleja, D Casso, W Chen, R Hatori, V Kumari, S Liu, K Makhijani, B Ng, N Ninov, JR Ortigão-Farias, S Bray, E Rulifson, JF de Celis, U Heberlein, K Irvine, J Reinitz, C Pitsouli, R Schüh, S Younger and M Affolter for suggestions, technical help, fly strains and reagents, the Developmental Studies Hybridoma Bank for antibodies, and the Bloomington stock center for fly strains.

## Additional information

### Funding

| Funder | Grant reference number | Author |
| --- | --- | --- |
| National Institutes of Health | GM030637 | Thomas B Kornberg |

The funders had no role in study design, data collection and interpretation, or the decision to submit the work for publication.

### Author contributions

PRR, LL, AG, SR, Conception and design, Acquisition of data, Analysis and interpretation of data, Drafting or revising the article; HH, Acquisition of data, Analysis and interpretation of data, Drafting or revising the article; TBK, Conception and design, Analysis and interpretation of data, Drafting or revising the article

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
