## [Decision Letter]

Thank you for submitting your work entitled "Developmental compartments in the larval trachea of *Drosophila*" for peer review at *eLife*. Your submission has been favorably evaluated by Diethard Tautz (Senior Editor), a guest Reviewing Editor, and three reviewers.

The reviewers have discussed the reviews (which are attached for information) with one another, and the Reviewing Editor has drafted this decision to help you prepare a revised submission.

The manuscript reports the existence of three regions of restricted lineage (compartments) in the trachea of the *Drosophila* larva: Dorsal trunk (DT), Dorsal branch (DB) and Transverse connective (TC). The identity of each of these compartments is determined with a specific selector gene and the boundaries between them are sites of Notch signaling. This is the first demonstration of the process of compartmentalization in a tubular internal structure.

Essential revisions:

The manuscript reports the existence of three regions of restricted lineage (compartments) in the second tracheal metamere (Tr2) of the *Drosophila* larva. Cell lineage experiments, using different marking methods, define three compartments: Dorsal trunk (DT), Dorsal branch (DB) and Transverse connective (TC). Marked clones initiated during the repopulation of the Tr2 metamere fail to cross well-defined borders between DT and DB and between DT and TC.

A key observation is that these same borders are independently defined by the expression of transcription factor encoding genes like *spalt, knirps* or *cut* that are specifically expressed in the DT, DB or TC. Conforming the classical selector gene hypothesis, these genes determine the identity of a specific compartment; alteration in their function (gain or loss) causes the corresponding alteration in compartment identity. This is reflected in the behaviour of, for example, *sal* mutant clones (Figure 5) that sort out from DT cells and acquire expression of *knirps* that determines DB identity.

Furthermore, the DT/DB and DT/TC boundaries delimit Notch expression and of its ligands Delta and Serrate, suggesting that Notch signaling is involved in the formation and maintenance of those borders, just like in the D/V border in the wing disc. This aspect is not sufficiently documented, as I point out below.

So far compartments in *Drosophila* have been demonstrated only in flat epithelia e.g. imaginal discs and abdominal histoblasts. This is the first demonstration of developmental compartments in an internal tubular organ. It is a significant piece of novel information that adds to the universality of the phenomenon of compartmentalization. It also implicates the existence of specific "genetic addresses" that program the development of various tracheal branches.

The technical quality of the paper is high. The conclusions are based in clear experiments and the results allow firm conclusions. The images supporting the conclusions are also clear and convincing, especially those referring to the demonstration of clonal restrictions and the expression data.

Considering all together, this work is potentially publishable in *eLife*, but some aspects remain unclear that preclude acceptance at this point. The authors should be asked to consider the comments below and to perform additional experiments and changes in the manuscript to meet those criticisms.

A serious problem concerns the role of Notch signaling, which is not clearly established. This is a criticism raised by all the reviewers. It is also a major issue, for one of the strengths of the work is that, just like in the wing imaginal disc, the lineage borders described here are associated with Notch signaling, which is presumably necessary for the formation and maintenance of those borders. The authors show the Notch function is necessary for normal morphogenesis of the dorsal branch, but the requirement of Notch signaling for the maintenance of the DT/DB or the DT/TC boundaries has not been investigated. The authors should perform additional experiments to address this problem. For example, as indicated by an expert reviewer, it should be possible to conduct a lineage experiment in trachea depleted of Notch function (*btl >NotchRNAi*) to check on the stability of the DT/DB border under those conditions.

Another problem is that the novelty of the results, that is, the functional connection between lineage restrictions, transcription factors and Notch signaling in building a tubular structure like the trachea, does not come out clearly in the manuscript. The authors should try to put their work in the context of what is known about the D/V border in the wing disc.

Minor points:

The authors state that they have screened 1300 enhancer trap lines, but do not mention any specific result. Which new tracheal expression patterns were identified in this screen? Several factors mentioned in the manuscript have already been reported in previous publications that should cited.

Reviewer #1:

The work submitted by Kornberg and colleagues describes the existence of lineage restriction boundaries in the developing larval tracheal system in *Drosophila*. Using clonal analyses, forward and reverse genetic, the existence of developmental compartments is convincingly demonstrated. The role of Notch signaling is investigated and shown to be important for boundary formation. This is the first demonstration of developmental compartments in a branched organ, and it will be interesting to find out whether other branched organs are also organized in a similar manner. The experiments are well done and presented.

Reviewer #2:

The submitted manuscript builds on the authors' previous work showing that cells in the second tracheal metamere (Tr2) of *Drosophila* larvae reenter the cell cycle during the 3rd instar stage (Guha and Kornberg, 2005; Guha et al., 2008). This repopulation is an example of organ renewal and cell division by differentiated cells. The authors have now further investigated this repopulation phenomenon. By conducting lineage analyses they found that the repopulating cells do not intrude into adjacent tracheal branches (lineage restriction). This fits with the notion that the branches behave as developmental compartments. Adjacent developmental compartments are distinguished by differences in the expression of particular genes, such as selector genes. The authors found that in Tr2, adjacent branches could indeed be distinguished based on the expression of such genes. Clonal depletion (e.g. Sal^-^) in compartments where they are expressed led to a sorting out from the neighboring cells and the acquisition of gene expression patterns seen in the adjacent compartment. Notch activity is highest in the DB cells at the branch sites. Depletion of Notch in trachea led to a morphological defect at the proximal DB. Taken together with other observations the authors propose that the Tr2 represent developmental compartments.

Compartments are a fascinating phenomenon in development biology. In *Drosophila*, the wing disc, a sac-like organ in which the compartments and their boundaries were discovered, has been used as a model organ for studying compartments. The authors provide evidence that Tr2 is an additional but distinct type of *Drosophila* organ (an internal tubular organ) that also has developmental compartments. This makes the manuscript potentially interesting for developmental biologists. However I hesitate to recommend publication of the manuscript in *eLife* in its current form due to the following reasons:

First, the relevance of the lineage restriction remains vague. The coincidence of lineage restriction and Notch signaling remains poorly investigated. Notch activity is high at branch points. Depletion of Notch in trachea caused a morphological defect at the proximal DB. The authors did not show how this influenced the lineage restriction. Is the lineage restriction at the junction of DB and DT still intact or have DB cells intruded the DT? The authors should conduct a lineage experiment in *btl >NotchRNAi* conditions.

Why is it worth studying Tr2? Per se it is interesting however the authors do not sufficiently highlight the potential value of the discovery. It is alluded to in the Discussion but the relevance, significance, and novelty of the results remain vague. Prefacing the relevance in the Introduction might help.

The authors could also compare and contrast the molecular mechanisms underpinning the formation and characteristics of developmental compartments in the DV boundary of the wing disc and the Tr2.

Second, the novelty of some of the results is questionable. The authors describe their analysis of gene expression patterns in the Tr2 as if they are the first ones to discover the different expression domains by screening enhancer trap lines. However, most of the expression patterns have already been described and the fact that the authors did not mention these papers (please see subheading “Delimited gene expression domains of the dorsal trunk, dorsal branch and transverse connective”) gives the wrong impression. This oversight should be corrected.

It is not clear from the information provided if the branch specific gene expression patterns also exist in other metameres in the larva (possibly at lower levels) or in embryos. If they are common, then the lineage restriction is due to intrinsic gene patterns of all tracheal metameres, and not a consequence of repopulation in Tr2. If the pattern is very specific, then an unknown, Tr2-specific signal might impose both lineage restrictions and repopulation, and such a finding could explain why Tr2 is repopulated. The authors should at least discuss this.

Third, as written, the manuscript will be very difficult to understand for non-developmental biologists.

Reviewer #3:

Developmental compartments have been well studied in the *Drosophila* imaginal discs and in the vertebrate nervous system, among other tissues. Regionalization depends on localized domains of gene expression (usually transcription factors) that confer compartment-specific identity. Signaling systems including Notch and Ephrin/Eph have been implicated in boundary formation and maintenance. Here, Rao et al explore the proliferation in the tracheal system, and provide evidence for a boundary of lineage restriction that coincides with gene expression domains. They present reasonable evidence that cells respect a boundary between dorsal trunk and dorsal branch that coincides with a domain of Spalt and Knirps expression, and some evidence that local activation of Notch signaling may be involved. Much of what is described is analogous to the roles of transcription factors and localized Notch activation in imaginal disc compartment systems, though much of this parallel is not explored in the text. The novelty here lies in the finding of these relationships within the tracheal system, in which the tubular epithelium has a different topology that the flat imaginal disc epithelia.

The data are generally clear-cut, and with the exception of the effects of manipulating Notch signaling produce results that are fully consistent with the authors’ model. The effects of manipulating Delta expression and Notch activity (Figure 6) appear less clear cut than those of manipulating Sal and Kni (Figure 5), but the authors conclude that activity is required at the DT/DB border. I am not entirely persuaded that the data on Notch signaling fully supports the authors claim that Notch signaling "was required for the identity and specializations of border cells".

---

## [Author Response]

*Essential revisions: […] A serious problem concerns the role of Notch signaling, which is not clearly established. This is a criticism raised by all the reviewers. It is also a major issue, for one of the strengths of the work is that, just like in the wing imaginal disc, the lineage borders described here are associated with Notch signaling, which is presumably necessary for the formation and maintenance of those borders. The authors show the Notch function is necessary for normal morphogenesis of the dorsal branch, but the requirement of Notch signaling for the maintenance of the DT/DB or the DT/TC boundaries has not been investigated. The authors should perform additional experiments to address this problem. For example, as indicated by an expert reviewer, it should be possible to conduct a lineage experiment in trachea depleted of Notch function* (btl >NotchRNAi) *to check on the stability of the DT/DB border under those conditions. Another problem is that the novelty of the results, that is, the functional connection between lineage restrictions, transcription factors and Notch signaling in building a tubular structure like the trachea, does not come out clearly in the manuscript. The authors should try to put their work in the context of what is known about the D/V border in the wing disc.*

We thank the reviewers and editors for the considerable effort that review of this manuscript demanded, and for the thoughtful suggestions. We are pleased to report that the major point that was raised – that we characterize the role of Notch signaling at the compartment border in Tr2 – has been resolved. This issue had been explored previously with such genotypes as the suggested *btl >Notch-RNAi*, but informative results had not been obtained. We explored several different genetic contexts, and as described in the revised manuscript, one was found that compromised the mutant clones so they did not sort out from their neighbors and did not respect the border (see Figure 6).

We addressed most of the minor points, but we are not able to provide more extensive descriptions of the enhancer trap screen as it was carried out many years ago and we lack sufficient documentation or access to the collection. Results from the screen directed us to genes that were characterized in this paper, but none of the lines were actually used for the studies that are reported.

As requested, Notch signaling at the wing disc D/V border is discussed and referenced and the text has been revised to reflect and highlight the new data. We have tried to reference all prior literature appropriately, but welcome any further suggestions from the reviewers.